# Is Hyperspectral Imaging Suitable for Assessing Collateral Circulation Prior Radial Forearm Free Flap Harvesting? Comparison of Hyperspectral Imaging and Conventional Allen’s Test

**DOI:** 10.3390/jpm11060531

**Published:** 2021-06-09

**Authors:** Diana Heimes, Philipp Becker, Daniel G. E. Thiem, Robert Kuchen, Solomiya Kyyak, Peer W. Kämmerer

**Affiliations:** 1Department of Oral–and Maxillofacial and Plastic Surgery, University Medical Center Mainz, Augustusplatz 2, 55131 Mainz, Germany; daniel.thiem@uni-mainz.de (D.G.E.T.); Solomiya.kyyak@unimedizin-mainz.de (S.K.); peer.kaemmerer@unimedizin-mainz.de (P.W.K.); 2Department of Oral and Maxillofacial Surgery, Federal Armed Forces Hospital, Rübenacher Straße 170, 56072 Koblenz, Germany; Becker-ph@web.de; 3Institute for Medical Statistics, Epidemiology and Informatics, University Medical Center of the Johannes-Gutenberg-University Mainz, 55131 Mainz, Germany; robert.kuchen@uni-mainz.de

**Keywords:** hyperspectral imaging, Allen’s test, radial forearm free flap, microvascular surgery, microsurgery, reconstructive surgery, perfusion monitoring, flap imaging

## Abstract

(1) Background: This cross-sectional study aims to compare a new and non-invasive approach using hyperspectral imaging (HSI) with the conventional modified Allen’s test (MAT) for the assessment of collateral perfusion prior to radial forearm free flap harvest in healthy adults. (2) HSI of the right hand of 114 patients was recorded. Here, three recordings were carried out: (I) basic status (perfusion), (II) after occlusion of ulnar and radial artery (occlusion) and (III) after releasing the ulnar artery (reperfusion). At all recordings, tissue oxygenation/superficial perfusion (StO_2_ (0–100%); 0–1 mm depth), tissue hemoglobin index (THI (0–100)) and near infrared perfusion index/deep perfusion (NIR (0–100); 0–4 mm depth) were assessed. A modified Allen’s test (control) was conducted and compared with the HSI-results. (3) Results: Statistically significant differences between perfusion (I) and artery occlusion (II) and between artery occlusion (II) and reperfusion (III) could be observed within the population with a non-pathological MAT (each <0.001). Significant correlations were observed for the difference between perfusion and reperfusion in THI and the height of the MAT (*p* < 0.05). Within the population with a MAT >8 s, an impairment in reperfusion was shown (each *p* < 0.05) and the difference between perfusion and reperfusion exhibited a strong correlation to the height of the MAT (each *p* < 0.01). (4) Conclusions: The results indicate a reliable differentiation between perfusion and occlusion by HSI. Therefore, HSI could be a useful tool for verification of the correct performance of the MAT as well as to confirm the final diagnosis, as it provides an objective, reproducible method whose results strongly correlate with those obtained by MAT. What is more, it can be easily applied by non-medical personnel.

## 1. Introduction

The “Chinese flap”, the fascio-cutaneous radial forearm free flap (RFFF) was first described in 1981 by Yang et al. [1,2]. It is used for diverse reconstruction purposes with a survival rate of 97% [3,4]; the popularity of its use has increased due to its pliability and thinness, the ease of flap raising using a two-team approach, a consistent anatomy, and the long and high-caliber vascular pedicle [1,2,3]. As the flap is vascularized by a segment of the radial artery that needs to be removed during the surgery, it is essential to ensure an adequate perfusion of the whole hand by the ulnar artery alone. Though, radial artery occlusion is a common complication with frequencies ranging from 1–33% [5]. Here, the Allen’s test, a simple bedside test is traditionally used as a preoperative assessment ahead of RFFF harvest. Its validity depends on the degree of arterial compression applied by the examiner, as well as the subjective evaluation of the reperfusion and a potential error by the hyperextension of the hand [3,6]. Furthermore, there are technical differences in the test procedure as well as the cut-off point for reperfusion (the shorter the cut-off time, the more sensitive and less specific Allen’s test is [7]). In addition, Allen’s test requires the cooperation of the patient for its correct performance and does not give information about the vascular anatomy [8]. As acute ischemia of the hand after flap raising has been reported, even after satisfactory Allen’s test results, further measurement methods have been evaluated to minimize the risk of such ischemic complications [3,4,6]. Here, arteriography is a very precise but also highly invasive method for assessing the vascular anatomy and perfusion of the donor hand. Color flow duplex scanning and doppler ultrasound are noninvasive options that require experience and rely on a subjective interpretation [3]. Besides, transcutaneous pulse oximetry has been shown to correlate with arterial doppler waveforms and therefore provides an objective means of monitoring of potential ischemic complications [9].

Hyperspectral imaging (HSI) is an imaging modality for medical applications and has been tested for the determination of perfusion parameters for diabetic foot and skin ulcer [10,11,12,13], tissue perfusion measurement and wound analysis [14,15] as well as flap monitoring [16]. TIVITA™ (Diaspective Vision, Pepelow, Germany) is a new, contact-free HSI system for the assessment of tissue oxygenation and perfusion. It is an internal pushbroom imaging spectrograph (CMOS camera) and acquires a three-dimensional HyperCube with spatial (x, y; resolution 0.1 mm/pixel at 50 cm distance) and spectral (λ; resolution 5 nm) dimensions [17]. Every point in a row (x-axis) is analyzed in parallel; the row is moved along the y-axis and the spectral dimension (λ) is generated [18,19]. The system detects hemoglobin and its derivates oxyhemoglobin, deoxyhemoglobin and water [20]. Optical remission spectroscopy in the visible and near infrared range (400–1000 nm) allows contact-free acquisition of information about tissue properties, such as tissue oxygenation/superficial perfusion (StO_2_ (0–100%); 0–1 mm depth), tissue hemoglobin index (THI (0–100)), near infrared perfusion index/deep perfusion (NIR (0–100); 0–4 mm depth) and tissue water index (TWI (0–100)) without influencing the tissue (Figure 1).

StO_2_ reflects the percentage of hemoglobin oxygen saturation in the capillary area of the tissue microcirculation and shows changes in tissue oxygenation. Thus, StO_2_ represents the tissue oxygen saturation, which is mainly due to the blood volume in the venous part (75%) of the microcirculation after the oxygen has been released to the surrounding tissue. The parameter NIR perfusion (near infrared) describes the quality of the blood flow, which is determined by the relative oxygen saturation of the hemoglobin and the relative hemoglobin content in the microcirculatory system in deeper tissue layers of 4 to 6 mm. This parameter can be used to identify undersupplied tissue areas in deeper layers. The color scale ranges from red (high perfusion) to blue (low perfusion). THI describes the existing hemoglobin distribution in the superficial microcirculatory system. By this means, index parameter problems with the arterial supply or the venous drainage can be recognized. The color scale ranges from red (high hemoglobin content) to blue (low hemoglobin content). As an index value, TWI describes the relative water content in the tissue area under consideration [21]. The tissue is irradiated with white light and the remitted light is detected; scattering and absorption by tissue structures depends on the wavelength [13,19] and light penetration depth is approximately 0.8 mm (500 nm) to 2.6 mm (1000 nm). The method has previously been described by Klucke et al. [17].

In the need of an objective, reliable and investigator-independent method to evaluate the vascular perfusion of the donor hand, this cross-sectional study aims to compare a new and non-invasive approach using HSI with the conventional Allen’s test for the assessment of collateral perfusion prior to RFFF harvest in healthy adults. The findings obtained in this study will be used to set limits for the evaluation of hyperspectral data, in order to facilitate interpretation and perioperative assessment.

## 2. Materials and Methods

### 2.1. Patients

The patients included within this study were picked randomly and independently of their medical history as part of the preoperative assessment ahead of surgery. All patients had a modified Allen’s test and a TIVITA™ (Diaspective Vision, Pepelow, Germany) scan as part of the assessment. The study was approved by the local ethic committee of Rhineland-Palatinate (registration number: 2020-15022_1) and was conducted in accordance with the protocol and in compliance with moral, ethical and scientific principles governing clinical research as set out in the Declaration of Helsinki of 1975 as revised in 1983.

### 2.2. Methods

All tests were performed by the same surgeon to avoid interobserver variability.

#### 2.2.1. Allen’s Test

Modified Allen’s test (MAT) was carried out as previously described by Habib et al. and Abdullakutty et al. (the patient makes a fist for 30 s, while pressure is applied on the ulnar and radial artery to occlude them. The patient opens the fist, and the ulnar artery is selectively released) [7,22]. An 8 s cut-off point was used to discriminate between positive test and results without pathological findings [7,23,24]. A longer period to arterial refill indicated a vascular anomaly potentially resulting in ischemia after radial artery harvest.

A time period of 9 s and above until reperfusion was evaluated as positive and therefore pathological; tests with reperfusion times of 0–8 s were considered negative and therefore non-pathological.

#### 2.2.2. HSI Imaging

The TIVITA™ Tissue System (Diaspective Vision, Pepelow, Germany) is an HSI system consisting of a hyperspectral camera, a lens, an illumination unit, a medical cart, a box-computer and the integrated TIVITA™ Suite basic software. The light source is arranged around the camera lens and consists of six halogen spots (20 W each). The standard measurement distance is 50 cm, represented by two indicator light points in an overlapped position. The patient was asked to place the hand flat on a table with the palm up. The central point for all measurements was the middle of the palm. After HSI images are recorded over 10 s, additional 8 s are needed to compute a RGD (red, green and blue) true colour image and additional four pseudo-colour images, representing the physiologic parameters. The camera-specific software package (TIVITA™ Suite) was used to quantify the generated information [20]. Three circular shaped regions of interest (ROI) that contain the mean value of the spectral and spatial information per pixel were manually positioned. The ROI were: in the middle of the palm (40 pixel), the proximal phalanx of the thump (15 pixel) and the proximal phalanx of the index finger (15 pixel). The software automatically calculated the average values for each perfusion parameter (StO_2_, THI, NPI and TWI). Afterwards, a mean of the three measurements was calculated (Figure 2).

The measurement was performed in a dark room with constant temperature. (I) First, a basis recording was conducted in order to show the individual hands’ perfusion (perfusion). (II) Then, MAT was carried out by occluding both the radial and the ulnar artery (occlusion). This was followed by a HSI measurement. (III) To verify the hand perfusion by the ulnar artery only, the radial artery stayed occluded while releasing the ulnar artery (reperfusion). Here, HSI-image acquisition for reperfusion started 1 s after release of the ulnar artery. For each time point the mean values of the following parameters were recorded: (StO_2_ (0–100%)), tissue hemoglobin index (THI (0–100)), near infrared perfusion index/deep perfusion (NIR (0–100)) and tissue water index (TWI (0–100)).

### 2.3. Statistics

In their review concerning the reliability and validity of the modified Allen’s test, Romeu-Bordas and Ballesteros-Pena listed a total of 9 studies that were included in the review. While the number of analyzed patients ranged from 42 to 150 (mean 81.88), the number of analyzed hands was on average 104.

Case number calculation (according to [25]):(1)$$n=K*\frac{\left[\left(R+1\right)\;-\;p_{2}*\;\left(R^{2}+1\right)\right]}{\left[p_{2}+{\left(1-R\right)}^{2}\right]}$$

The calculation is based on the study by Grambow et al. [26] and the measured differences in tissue oxygen content, which were found to be significantly different.
(2)$$n=7.85*\frac{\left[\left(0.757+1\right)\;-\;0.7*\;\left(0.757^{2}+1\right)\right]}{\left[0.7+{\left(1-0.757\right)}^{2}\right]}=124.578$$

Considering the previous studies and the sample size calculation, the numbers of cases were averaged, resulting in a total necessary volume of 114 patients.

In order to test the assumption that the measured values follow a normal distribution, a Kolmogorov–Smirnov and a Shapiro–Wilk test were previously performed. Correlation analyses were performed using Pearson and Spearman test. Furthermore, Wilcoxon tests were used in order to assess whether the population mean ranks differ between related samples (non-parametric statistical hypothesis test). To measure the strength of the relationship between two variables in the statistical population, the effect size was calculated according to the following formula:(3)$$r=\;\left|\frac{Z}{\sqrt{n}}\right|$$

The following definitions were used:0.1≤r<0.3~weak effect
0.3≤r<0.5~medium effect
r≥0.5~strong effect

Values are displayed as mean and standard deviation; where appropriate confidency intervals are given. Moreover, in addition to the measured values, a fictive value was calculated to reflect the return-to-normal perfusion. This was done as follwos:(4)$$RTP=100-\left(\frac{TP3}{TP1}*100\right)$$
with *RTP* and *TP* denoting return to perfusion time point, respectively. Statistical analyses were performed using SPSS version 24 for Windows (IBM, Armonk, NY, USA); A *p*-value *≤* 0.05 was termed significant.

## 3. Results

A total of 114 patients were included within this study, all of whom had a modified Allen’s test and an HSI-scan as part of the assessment. The results were categorized into two groups according to the results of the MAT. Non-pathological results were those with a time to reperfusion of less than 9 s; if it took 9 or more seconds, the MAT was considered pathological.

### 3.1. Population with a Non-Pathological Modified Allen’s Test

#### Allen’s Test

Here, 100 patients were included. Mean time to arterial refill was 4.12 s (SD = 1.903 s; 95% CI:3.74–4.50 s). The values were not normally distributed (Figure 3).

### 3.2. Hyperspectral Imaging

#### 3.2.1. Tissue Oxygenation (StO_2_)

The Shapiro–Wilk test was not able to reject the hypothesis that measured values at every time point were normally distributed. At time point I, mean StO_2_ was 51.34% (SD = 7.972%; 95% CI:49.76–52.92%). After vessel occlusion (II), mean StO_2_ decreased to 40.56% (SD = 6.929%; 95% CI: 39.19–41.93%) and after releasing the ulnar artery (III), StO_2_-values increased again up to 50.42% (SD = 8.117%; 95% CI: 38.81–52.03%) (Figure 4 and Figure 5). Pearson and Spearman test showed a statistically significant correlation between the different time points (each *p* < 0.001). Whereas mean ranks differed significantly between related samples time points I and II as well as between II and III (each *p* < 0.001), the values between time point I and III did not differ significantly (*p* = 0.06). This indicates a strong effect within the compared groups that showed significantly different mean ranks (r each >0.5) and a weak effect for group comparison I and III (r = 0.133).

#### 3.2.2. Near Infrared Perfusion Index (NIR)

Once again, a Shapiro–Wilk test did not significantly contradict that measurements during occlusion and reperfusion were normally distributed (*p* > 0.1). The values measured at time point I (both arteries open), however, do not seem to follow a normal distribution (*p* = 0.031). At time point I, mean NIR was 55.11 (SD = 7.236; 95% CI: 53.67–56.55). After occlusion of both vessels, the perfusion decreased (Mean = 48.14 ± 5.946; 95% CI: 46.96–49.32) and increased up to 53.88 (SD = 7.876; 95% CI: 52.32–55.44) after release of the ulnar artery (Figure 4 and Figure 5). Both, Pearson and Spearman correlation test showed statistically significant correlations between the different time points (each *p* < 0.001). Using Wilcoxon test, a significant difference between the distribution at the different time points could be demonstrated. Whereas for time points I and II as well as for II and III a strong effect (r > 0.5) was shown, between group I and III merely a weak effect was observed (r = 0.241).

#### 3.2.3. Tissue Hemoglobin Index (THI)

Whereas a Shapiro–Wilk test could not reject that the measured values of both, perfusion and reperfusion were normally distributed (*p* > 0.05), the values obtained during occlusion do not seem to follow a normal distribution (*p* = 0.005). Mean THI was 35.39 ± 7.661 (95% CI: 33.87–36.91) while arterial blood flow was present, whereas THI decreased to 20.52 ± 8.973 (95% CI:18.74–22.20) after occlusion of the arteries. After release of the ulnar artery, mean THI increased up to 38.52 ± 9.517 (95% CI: 36.63–40.41) (Figure 4 and Figure 5). Spearman and Pearson test showed statistically a significant correlation between all time points (*p* < 0.001). A significant difference between the mean ranks of the different time points could be shown by the Wilcoxon test. For time points I and II as well as for II and III a strong effect (r > 0.6) and between I and III a moderately strong effect could be shown.

#### 3.2.4. Tissue Water Index (TWI)

Whereas the Shapiro–Wilk test indicated that for both, time points II (both arteries occluded) and III (only radial artery occluded), the normality assumption could not be rejected, this was not the case for the baseline measurement. Mean TWI was 41.28 (SD = 3.690; 95% CI: 40.55–42.01) at rest and increased to 45.68 ± 3.959 (95% CI: 44.89–46.47) when occluding both vessels. After release of the ulnar artery, TWI decreased to 42.01 (SD = 4.162; 95% CI: 41.18–42.84) (Figure 4 and Figure 5). Both Spearman and Pearson correlation test showed statistically significant correlations between all time points (*p* < 0.001). Analyzing the difference in mean ranks, there were statistically significant differences between all groups corresponding to strong effects within the compared groups I and II as well as for II and III (r each > 0.5) and a weak effect for group comparison I with III (r = 0.231).

#### 3.2.5. Return-to-Perfusion Measurement

The return-to-perfusion (RTP) value indicates the difference in percent between the measurements at time point III and time point I. Shapiro–Wilk and Kolmogorov–Smirnov could not reject a normal distribution for StO_2_-RTP values. The mean difference for StO_2_ was 1.46% (SD = 8.84%; 95% CI: −0.29–3.21%), for NIR 2.27% (SD = 5.74; 95% CI: 1.13–3.41%), for THI −10.21% (SD = 21.79%; 95% CI: −14.53– -5.88%) and for TWI −1.82% (SD = 5.63%; 95% CI: −2.93– -0.7%) (Figure 5). Correlation analysis between the MAT and RTP-measurements by Pearson and Spearman test showed no significant correlation between the MAT and RTP-values for StO_2_ and NIR, whereas there was a statistically significant correlation between the MAT and THI values (*p* < 0.05) in the Spearmen test. On the other hand correlation analysis did not show a correlation between MAT and TWI.

### 3.3. Cases with Impaired Perfusion

#### 3.3.1. Allen’s Test

A total of 14 patients had a MAT with a time to reperfusion of longer than 8 s. In three cases, time to reperfusion was 9 s, in four cases 10 s, in two cases 11 s, in one case 14 s and in four cases, there was no reperfusion detectable after more than 20 s, which is why the test was terminated in such cases (termed as AT max.).

#### 3.3.2. Tissue Oxygenation (StO_2_)

A normal distribution of all time points analyzed within this study was detected. A statistically significant correlation between the different time points could be observed (each *p* < 0.05). Mean ranks differed significantly between the related samples time points I and II (*p* < 0.001) as well as between I and III (*p* = 0.048), whereas the values between time point II and III did not differ significantly (*p* = 0.076). This corresponded to a strong effect within the compared groups I and II (r = 0.6) and a moderately strong effect for group comparison I and III and II and III (r > 0.3).

#### 3.3.3. Near Infrared Perfusion Index (NIR)

A normal distribution for the measured values of all time points was seen. A statistically significant correlation between the different time points could be shown by Pearson and Spearman test with *p* < 0.01. There were significant differences in mean ranks between the related samples time points I and II as well as I and III (*p* = 0.002), whereas the values between time point II and III did not differ significantly (*p* = 0.213). This corresponded to a strong effect between the compred groups that showed significantly different mean ranks (r each > 0.5), whereas for group II and III only a weak effect could be shown (r = 0.235).

#### 3.3.4. Tissue Hemoglobin Index (THI)

The hypothesis of a normal distribution could not be rejected for time points I and III. Correlation analysis by Pearson and Spearman test could demonstrate a statistically significant correlation between the different time points with *p* < 0.001. Interestingly, statistically significant differences between mean ranks of time points I and II as well as II and III were shown, whereas time points I and III did not show such significant differences (*p* = 0.133). This corresponded to a strong effect between the compared groups that showed significantly different mean ranks (r each > 0.5), whereas for group I and III, a weak effect could be shown only (r = 0.284).

#### 3.3.5. Tissue Water Index (TWI)

For time points II and III, normally distributed values were observed. Both, Spearman and Pearson correlation test were able to show a statistically significant correlation between all time points. Mean ranks differed significantly between all groups, corresponding to a strong effect between time point I and II (r = 0.626) and a moderate effect between the other time points with r > 0.4.

#### 3.3.6. Return-to-Perfusion Measurement

Shapiro–Wilk and Kolmogorov–Smirnov tests could not reject the hypothesis of normal distribution for StO_2_, NIR, THI and TWI. There was a statistically significant correlation between the RTP-value and the MAT-measurements for StO_2_ (*p* < 0.001), THI (*p* = 0.004) and TWI (*p* = 0.011), whereas no significant correlation could be shown for NIR-index (*p* = 0.179) (Figure 6).

### 3.4. Patient Case

A 63-year-old patient presented with an oral mucosal lesion of the right floor of the mouth that had been present for a year. Clinically, there was an ulcerating lesion of about 2 cm. During staging, no further suspicions lesions could be detected; radiologically, neither lymphatic, nor osseus metastases were found. The interdisciplinary head and neck tumor board recommended resection of the tumor and bilateral neck dissection. To cover the defect, we planned to harvest a RFFF. However, the MAT showed a very poor perfusion of the right arm with a reperfusion time of over 20 s; the results could be confirmed by HSI. When the MAT was performed of the left arm, a time to reperfusion of 11 s was measured. However, the HSI measurement showed adequate perfusion with a satisfactory RTP-value for StO_2_, NIR, THI, and TWI (Figure 7). Considering the measurable reperfusion, the decision was made to harvest a RFFF from the left arm. With constant monitoring of the oxygen saturation by means of pulse oximetry, the RFFF could be harvested without complications. In the postoperative follow-up, the graft was adequately perfused and healed well (Figure 8).

## 4. Discussion

The radial artery is located in the lateral intermuscular septum between the brachioradialis and flexor carpi radialis muscles [1]. Entering the hand, the radial artery gives rise to the princeps pollicis artery and radial indices artery. The deep palmar arch is formed by the dorsal radial artery and the deep branch of the ulnar artery. Four palmar metacarpal arteries arise from the deep palmar arch and converge with the common palmar digital arteries. The superficial palmar arch is formed by the ulnar artery and the superficial branch of the radial artery with four common palmar digital arteries arising from the arch. The common digital arteries then divide into two proper palmar digital arteries [22,27,28] (Figure 9).

The vascular abnormality leading to impaired perfusion after raising the RFFF is a combined condition of an incomplete ulnar arterial supply to the hand and a missing communication between deep and superficial palmar arch. Coleman and Anson reported 12% of specimens to show a combination of the two abnormalities [29]. Strauch et al. reported the superficial arch and the deep arch to be incomplete in 21% and in 3% of cases, respectively. Other studies found the collateral circulation to be absent in 2% up to 20% of cases [28]. Complete superficial arches occur in 84% up to 90% of cases [4,27,30,31,32].

In this regard, a higher incidence of pathological results in Allen’s test could be expected. In a large study with 1000 patients undergoing cardiac catheterization, 49% had a normal Allen’s test (cut-off time <5 s). The authors classified 24% as borderline (5–9 s) and 27% as abnormal (>10 s) [22]. In 1990, Hosokawa et al. showed 5.8% of 1470 patients examined within the hospital to have an abnormal Allen’s test (time until recover of color 5 s). Unilateral abnormality was observed in 4.4%, bilateral abnormality in 1.4% of cases [2]. The incidence of an abnormal test increased with age (incidence of abnormality >80 years 6.9%). Since the average age of developing oral cancer is 62 years [33], a more accurate and safer test method is needed to ensure adequate blood supply by the ulnar artery when harvesting RFFF.

Nuckols et al. reported a sensitivity of 65% and a specificity of 76% (positive predictive value (PPV) = 93%, negative predictive value (NPV) = 35%) for Allen’s test with a cut-off time of 5 s [28]. This corresponds to the results of Husum et al. who indicated a NPV of 0.992, which yields a false-positive rate of to a 0.8% (1/100 hands with normal Allen’s test and an inadequate collateral circulation) [28]. According to this study, a normal test result would incorrectly indicate inadequacy in about 50% of cases [28]. A systematic review by Romeu-Bordas et al. evaluated the reliability and validity of Allen’s test in patients prior to radial artery puncture. They concluded Allen’s test to show inadequate diagnostic validity for screening deficits in the collateral circulation. Because of this, Allen’s test is termed to be no adequate predictor of hand ischemia: “Therefore, Allen’s test should not be systematized prior to performing an arterial puncture as an isolated screening test for collateral arterial circulation deficits of the hand and should not be considered an absolute contraindication for performing a transradial puncture presenting an abnormal result in the Allen’s test” [8]. False-negative Allen’s test could result in hand ischemia and necrosis, whereas false-positive test results cause a change of primary treatment plan, possibly resulting in a suboptimal therapy situation [4]. Initially, ischemic hand complications (IHC) appear in the form of pallor and progressive darkening of the skin. Chronic complications include pain, cold intolerance, ulceration, tissue necrosis and gangrene of the digits [4].

Therefore, a secure method for vascular assessment is needed. In daily routine, some supplemental properties are necessary to make the test feasible: the test should be noninvasive, fast as well as easy to perform and the evaluation needs to be objective and reproducible. Furthermore, the method needs to have a good predictive ability with a high sensitivity, specificity and accuracy. In this study, HSI was shown to detect perfusion deficits during MAT. HSI provides both, topographical and spectral information in an objective, reproductive and measurable manner. Combinations of values allows drawing conclusions about tissue perfusion. High THI and low StO_2_ indicates venous congestion, whereas low THI and low StO_2_ points to an arterial occlusion. A high NIR and a low StO_2_ indicated that deep tissue perfusion is given whereas superficial layers are undersupplied, whereas the contrary case points to a critical situation as superficial supply can clinically hide saturation problems in deeper tissue layers [21]. Moreover, if the reliability of Allen’s test as a screening tool is shown by a high number of successful and complication-free radial forearm free flap transfers, the remaining percentage of hand ischemia in the presence of non-pathological test results suggests the need for a more secure measurement method to increase patients’ safety.

The results of this study indicate that in patients with a non-pathological MAT (blood-refill time of less than nine seconds) all parameters collected during hyperspectral imaging significantly differ between both, the baseline measurement and the measurements taken during complete artery occlusion as well as between the measurement after release of the ulnar artery and during artery occlusion. Furthermore, THI-RTP values correlated with the MAT results. Hence, it can be concluded that the calculated ratio between the HSI-measurements at the beginning and the end of the test (RTP value) representing the ability to full reperfusion by only the ulnar artery reflects the time to reperfusion measured during MAT.

On the one hand, this indicates a reliable differentiation between perfusion and occlusion–as already shown by Grambow et al. in a rat in vivo model [26]–as well as the confirmation of non-pathological MAT. Therefore, the use of hyperspectral imaging for additional diagnostics in combination with MAT would be a useful tool to verify the correct performance of the test (differentiation of occlusion and perfusion) as well as to confirm the final diagnosis of a non-pathological MAT. With the aid of the RTP-value, a correlation between hyperspectral imaging and MAT could further be supported. We could not only show a safe differentiation between perfusion and occlusion status during a non-pathological MAT, but the system can also detect a pathological reperfusion based on the different parameters. This can be observed by the no longer significant differences between time points I and II as well as between II and III, but–due to the impaired reperfusion at time point III–significant differences between perfusion (time point I) and occlusion (time point II) as well as between perfusion (time point I) and reperfusion (time point III). Interestingly, we could show a statistically significant correlation between the RTP-value and the MAT results. This is in accordance with the observed outliers when comparing the hyperspectral data of patients with a non-pathological MAT and those with a pathological MAT. Here, it is clearly shown that the hyperspectral measurement values that are far beyond the norm correlate well with certainly pathological MAT values (AT max.).

Due to the overall group size and especially the rather small number of pathological MAT readings in our study, it was not possible to define safe cut-off values, rendering the fitting of neural-networks impossible. Nonetheless, the trends observed in this study indicate a potential use of hyperspectral imaging for reperfusion analysis thus offering an alternative or supplementary method to the gold standard.

Based on current data, an advantage of HSI over clinical assessment alone has already been observed for perfusion monitoring of microvascular anastomosed grafts, particularly by inexperienced personnel. Compared to visual assessment of the hand reperfusion during MAT, HSI offers some clear advantages as it provides an objective, reproducible method also feasible for non-medical personnel that has no interobserver error and–in contrast to the MAT–gives a visual and measurable feedback in case of insufficient artery occlusion or potential test error due to palmar hyperextension.

HSI is used both clinically and experimentally for numerous indications: In anesthesia and ICU, the technique is already used in critically ill patients to monitor micro- and macro-circulatory changes, tissue perfusion and oedema formation to reduce the negative effects of hemodynamic incoherence. Prior to HSI, skin mottling and capillary refill time were used to assess hemodynamic parameter, but, as in MAT, the inter-observer variability demonstrated contradictory findings and a variety of cut-off values were suggested [34]. In vascular surgery, HSI is used to provide objective decision criteria for determining the extent of amputation and to make predictions about the chance of healing of the amputation wound [35]. What is more, HSI has been frequently used in monitoring perfusion in microvascular anastomoses, both in experimental setups and in the clinical practice. Here, a clear advantage over the visual assessment of the grafts could be shown in numerous studies [21,26,36]. Similar results could also be shown in transplant medicine. Here Sucher et al. proved that by means of HSI it is possible to objectively assess whether an organ (the kidney) is suitable for transplantation even before the surgery. In addition, it is possible to check immediately after the transplantation whether the blood vessels have been sutured correctly and the organ is sufficiently supplied with blood. Until now, this was also decided by visual assessment, so the new HSI technique offers a clear advantage [37]. In visceral surgery, HSI is used for blood flow analysis, for example, to determine the extent of resection in the case of mesenteric ischemia, but also to assess anastomoses, as well as tubular gastric blood flow in esophageal resections [38,39]. At the current time, numerous other applications of the hyperspectral camera have already been developed. In addition to special products for the analysis of wounds and soft tissue, camera systems for the operating room and an endoscopic version have also been developed. For the resection of brain tumors, MRI scans are taken preoperatively in order to locate the tumor and subsequently plan the surgical intervention. Intraoperatively, there is currently a lack of tools to locate the tumor with certainty. Data to date indicate that it will most likely be possible to identify the tumor and resect it safely using this method [40]. This large number of potential application areas and the already established use in everyday clinical life point to a great benefit of hyperspectral technology in the future. In particular, subjective assessment criteria can be replaced by means of this technique and thus the therapy of patients can be improved.

As the ambient light conditions affect the parameter values, cautious interpretation is demanded. To the authors knowledge, this is the first study assessing the feasibility of HSI to collateral circulation prior to RFFF harvest. Regarding the number of cases within this study, the present results need to be classified as of descriptive nature. Further clinical studies must be conducted in order to set cut-off values indicating a save arterial refill during HSI assisted MAT. In general, perfusion markers should return to the level of the baseline measurement after releasing the ulnar artery. If the parameters stay low, an adequate perfusion is not guaranteed. What is more, in addition to the measurements performed during this study, further measurements at later time points could have been performed to assess the long-term which could have provided further insight. However, in this study, further measurements were deliberately omitted because we aimed to (1) to ensure the greatest comparability possible between the gold standard and the new method and (2) develop a new method for everyday clinical use. This method should be as simple, fast, and objective as possible, so further, time-consuming follow-up measurements were not necessary. With this study, we aimed to investigate the new method exactly as it should be applied in clinical practice. Despite the more limited assessability of long-term perfusion due to the reduction in the number of measurements, we were able to demonstrate that this method can clearly distinguish between appropriate, questionable, and non-appropriate donor sites. Nevertheless, especially in patients with impaired perfusion (questionable and non-appropriate), the assessment of tissue perfusion in later recordings would be of interest. A clear distinction could be made between delayed complete reperfusion and lack of reperfusion, thus identifying patients who would be expected to develop long-term tissue damage.

As the assessment of collateral perfusion by the ulnar artery is not just mandatory in the field of flap raising, but also for arterial puncture in anesthesia, coronary artery intervention and bypass, there is a high need for such a technique.

## 5. Conclusions

This study was able to show that, using HSI, a safe differentiation between perfusion and occlusion is possible and that this method has a good correlation to the present gold standard, the MAT. Therefore, HSI could serve as an additional method for the assessment of the collateral circulation of the hand prior to invasive interventions involving the damage or harvesting of radial artery. HSI provides some advantages over the single visual assessment, as it provides objective, reproducible results without interobserver error, can also be applied by non-medical personnel, and gives a visual and measurable feedback. Yet, limitations are imposed by the poor data situation and examination-related measurement errors.

## Figures and Tables

**Figure 1 jpm-11-00531-f001:**
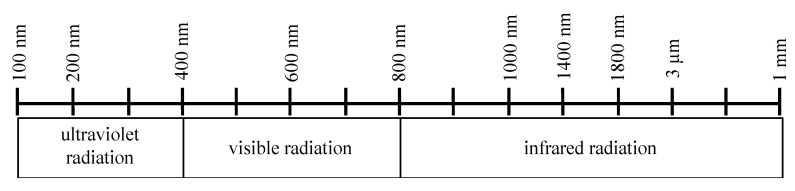
Wavelengths of different types of light. The hyperspectral camera processes visual light with a wavelength from 380 to 740 nm and light in the near infrared range from 750 to 1000 nm.

**Figure 2 jpm-11-00531-f002:**
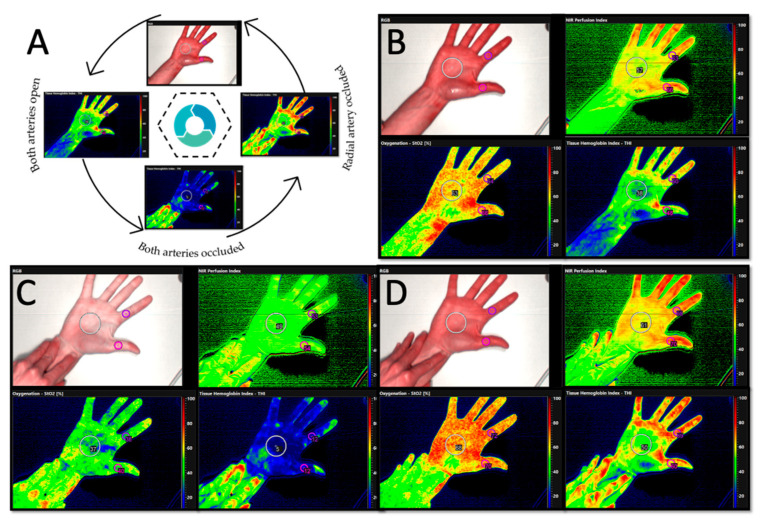
(**A**) Course of the experiment. First, a baseline measurement was taken (time point I, both arteries open), then both, the radial and the ulnar artery were occluded (time point II) and at time point III, the ulnar artery was released. Hyperspectral images were taken at all three time points throughout the course of the experiment as part of the hyperspectral accompanied MAT. In (**B**) the baseline measurement is displayed showing the NIR perfusion index, oxygenation and tissue hemoglobin index. Three circular shaped regions of interest (ROI) were manually positioned. (**C**) After occlusion of the ulnar and radial artery, the hand obviously turns pale. The values for NIR perfusion index decrease from 57 to 49 in the middle of the palm indicated by a less red and more green color of the hand. The same applies for the tissue oxygenation showing many red spots before occlusion and a green and blue color after occlusion of the vessels (StO_2_ decrease from 63% to 37%). Changes in THI are most obvious indicated by a color change from green to dark blue (THI decrease from 38 to 5). (**D**) shows the measurements after release of the ulnar artery in a healthy participant. The return to perfusion is indicated by the red and green colors in all pictures. After reperfusion, there is a slight hyperemia of the hand, which is evidenced by the increase in the measured values (NIR1: 57 and NIR3:61; StO_2_1: 63% and StO_2_3: 68%; THI1: 38 and THI3:50).

**Figure 3 jpm-11-00531-f003:**
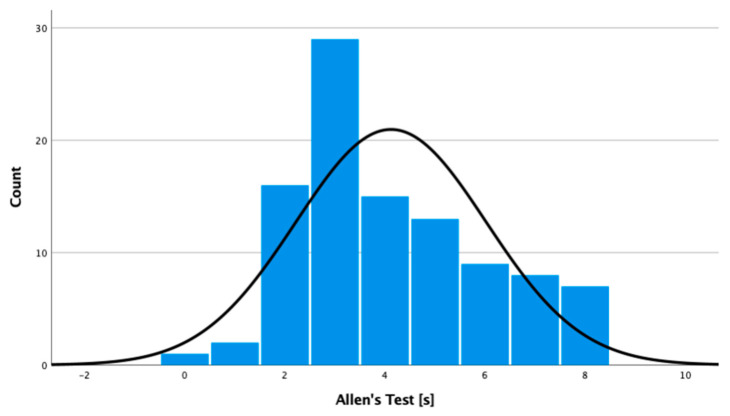
Boxplot diagram of the values obtained from the population with a non-pathological modified Allen’s test.

**Figure 4 jpm-11-00531-f004:**
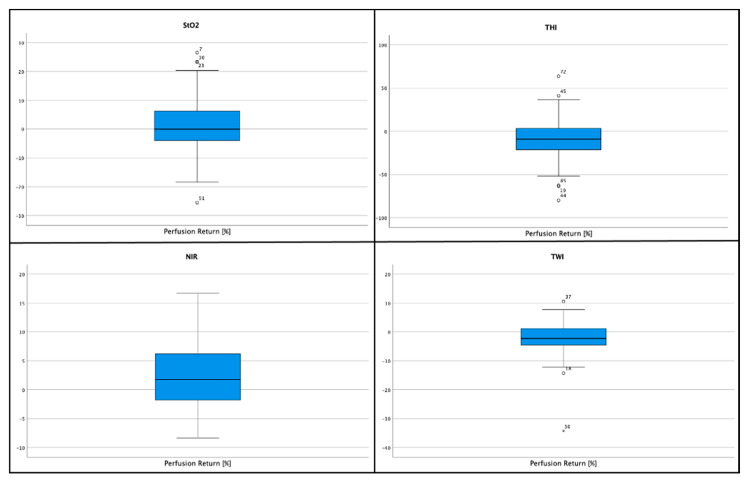
Boxplot diagrams for return-to-perfusion measurements for StO_2_, NIR, THI and TWI values obtained from the population with a non-pathological modified Allen’s Test.

**Figure 5 jpm-11-00531-f005:**
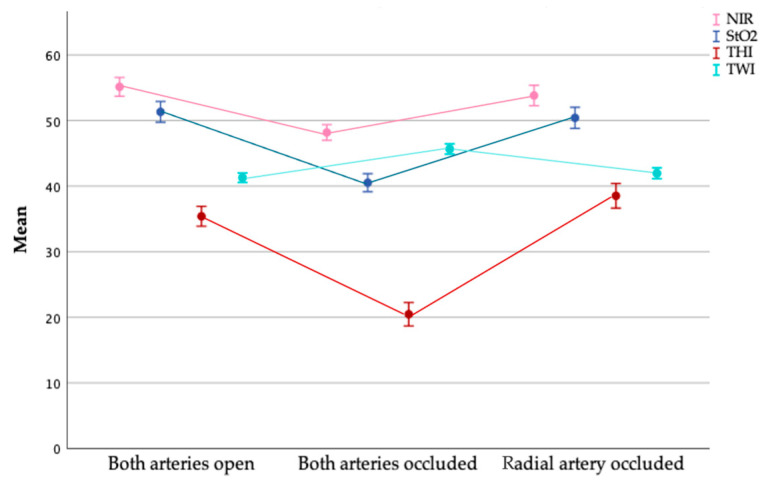
Measurements of the hyperspectral analysis over the time course of the experiment. The measured value at time point I symbolizes the baseline measurement, at time point II the values under occlusion of both vessels and at time point III the measured values after opening the ulnar artery. The NIR values (near infrared index/deep perfusion) are shown in pink, the StO_2_ values (superficial perfusion) in blue, the THI (tissue hemoglobin index) in red and the TWI (tissue water index) in cyan.

**Figure 6 jpm-11-00531-f006:**
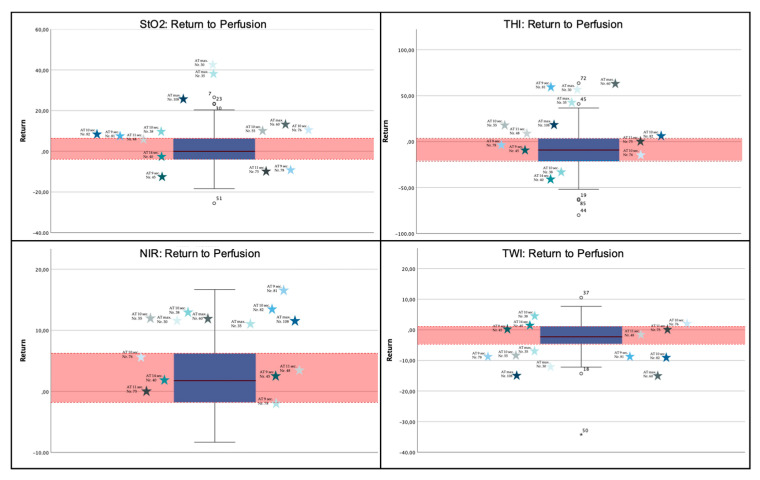
Boxplot diagram of the Return-to-Perfusion (RTP) measurements. The red box highlights the area between the 1st and 3rd quartile. The stars indicate the different subjects’ measurements that had a MAT >8 s. As can be observed, some values lie between the 1st and the 3rd quartile, whereas others are located within the 4 inter quartiles range (IQR). Others lie due to their extreme values beyond these areas and are therefore considered outliers. With few exceptions, the measurements defined as outliers belong to the same patients. As can be seen, RTP-values for StO_2_, THI and TWI of patients with a clear pathological MAT typically lie beyond the 4 IQR (Nr. 30, 35, 60, 108). Others, with a MAT of 9–15 s–depending on the definition–rated as pathologic or non-pathologic, typically fall within the 4 IQR. The allocation of the values on the X-axis has no meaning.

**Figure 7 jpm-11-00531-f007:**
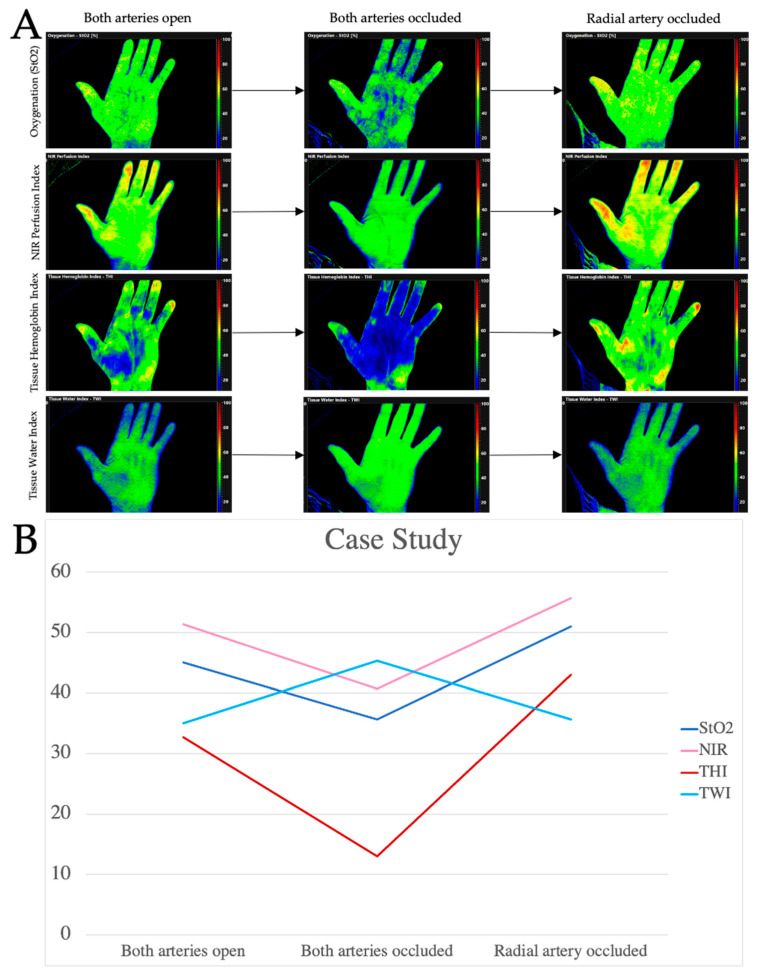
In (**A**) the measurements at the three different time points are shown. A baseline measurement was taken (time point I, both arteries open), then both, the radial and the ulnar artery were occluded (time point II) and at time point III, the ulnar artery was released. At time point I, tissue oxygenation (StO_2_) was 45%. After occluding the arteries, the values decreased to 35.67% and increased again after release of the ulnar artery (51%) as can be seen in (**B**). This observation is consistent with the measurements of the NIR perfusion index (NIR) showing values of 51.33 at time point I, 40.67 at time point II and 55.67 at time point III (see B). The image shows very impressively the trend of the hemoglobin index (THI) during the course of the experiment. With both arteries open, the THI was 32.67; after occlusion of both arteries, the values decreased to 13 and increased again after release of the ulnar artery (43). As can be seen in the images in A, tissue water index (TWI) increased during artery occlusion showing values of 45.33 while both, the baseline measurement and the measurement after release of the ulnar artery showed values of 35. The Return-to-perfusion (RTP) value indicates the difference in percent between the measurements at time point III and time point I. Here, RTP-values for StO_2_ were −13.33%, for NIR −8.46%, for THI −31.62% and for TWI −1.91%. Those values corresponded to a strong return to perfusion after release of the ulnar artery showing a save perfusion of the hand by the ulnar artery alone.

**Figure 8 jpm-11-00531-f008:**
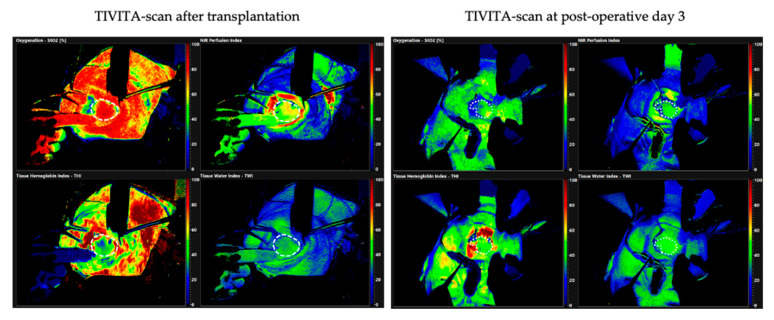
A TIVITA scan immediately after transplantation of the RFFF into the floor of the mouth is shown on the left. The circular mark (white dotted line) indicates the graft inside the patient’s mouth. The measurement shows a (typical) initially very high oxygenation of the graft with correspondingly high NIR values and adequately low THI values. In the course of time, up to postoperative day 3 shown on the right, a reduction of the StO_2_ and NIR values and a homogenization of the measured values over the area of the graft are visible. Based on previous studies [21], it could be determined that this process is typical for the proper healing and perfusion of the graft.

**Figure 9 jpm-11-00531-f009:**
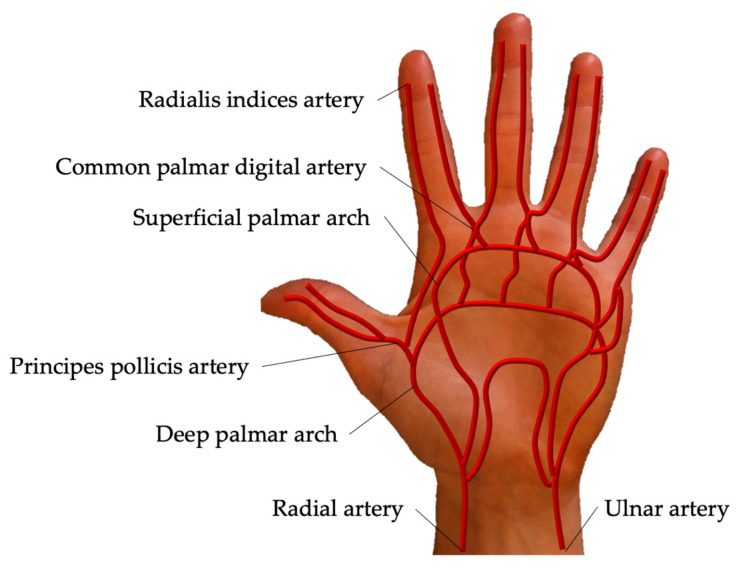
Palmar arch of the right hand. Entering the hand, the radial artery gives rise to the princeps pollicis artery and radial indices artery. The deep palmar arch is formed by the dorsal radial artery and the deep branch of the ulnar artery. The superficial palmar arch is formed by the ulnar artery and the superficial branch of the radial artery.

## Data Availability

The data supporting the conclusions of the article is included within the article. The raw data analyzed during the current study is available from the corresponding author on reasonable request.

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
