# Peer review of "Is Hyperspectral Imaging Suitable for Assessing Collateral Circulation Prior Radial Forearm Free Flap Harvesting? Comparison of Hyperspectral Imaging and Conventional Allen’s Test"

_jpm, 2021, doi:10.3390/jpm11060531_

Round 1
Reviewer 1 Report
The need of a more objective test method for a preoperative assessment ahead of RFFF harvest for the hand perfusion is described plausibly.
Regarding the technical and application features as well as the proven potential information content, hyperspectral imaging (HSI) seems to be an adequate method to become such a method.
As described in the manuscript, the drawbacks of Allen’s Test are
- the uncontrolled degree of arterial compression;
> HSI-parameter at TP_I in relation to TP_II can be a reliable measure for the correct performance of the occlusion - the subjective evaluation of the reperfusion; this parameter is in MAT assessed only by the visual impression of the recurrence of the color; this very subjective assessment says something about the reperfusion dynamics, but not necessarily about the reached perfusion quality;
> the HSI-measurement at the time TP_III, which is not exactly defined in the manuscript, provides objective parameter in comparison to TP_II and TP_I and could replace the MAT reperfusion time; is the measurement performed at approx.. 8 sec after the reperfusion started? The measurement needs at least 10 sec up to the results, thus a measurement of the reperfusion dynamics is difficult, but why further measurements were not performed to achieve more information about the final quality of the perfusion and possible errors by hyperextension? - that there is no information about the vascular anatomy;
> with the HSI-parameter an additional (uncomplete) differentiation between superficial (STO2, THI) and deep perfusion (NIR) is provided and opens up the possibility to analyse perfusion problems more differentiated. Could this be used for instance with respect to ischemic complications?
The results presented in the manuscript show an objective and significant differentiation between the baseline (TP_I) and the occlusion phase (TP_II) as well as the reperfusion phase (TP_III). The quantitative assessment of the last parameter in relation to the first is actually not sufficiently performed due to the restricted amount of data.
Generally, the comparison of a suboptimal test method, which should be replaced, but is actually the gold standard, with a potentially better and more objective method is difficult, and finally the value of the new method can only be validated with larger studies and by means of the clinical results.
Author Response
This is the revision of the manuscript entitled “Is hyperspectral imaging suitable for assessing collateral circulation prior radial forearm free flap harvesting? Comparison of hyperspectral imaging and conventional Allen’s test“
We would like to thank Reviewer 1 for the comprehensive review and summary of the study results, which show the extensive review of our article. From this we have extracted the following comments. We hope that this interpretation corresponds to what Reviewer intended.
Comment #1:
The HSI-measurement at the time TP_III, (A) which is not exactly defined in the manuscript, provides objective parameter in comparison to TP_II and TP_I and could replace the MAT reperfusion time; is the measurement performed at (B) approx. 8 sec after the reperfusion started? The measurement needs at least 10 sec up to the results. Thus, a measurement of the reperfusion dynamics is difficult, but why further measurements were not performed to achieve (C) more information about the final quality of the perfusion and possible errors by hyperextension?
Answer #1:
(A) HSI-image acquisition for TP_III started 1 second after release of the ulnar artery (see lines 147, 148). (B) After HSI images are recorded over 10 s, additional 8 s are needed to compute a RGD (red, green and blue) true colour image and additional four pseudo-colour images, representing the physiologic parameters (see Bahkatov et al. 2005, Optical properties of human skin, subcutaneous and mucous tissues in the wavelength range from 400 to 2000 nm) (see lines 132–135). (C) As noted, additional insight regarding the long-term tissue perfusion could have been gained by later hyperspectral analyses of the patient's hand. However, in the present study, further measurements were deliberately omitted because the aim of this work was (1) to make the new measurement as comparable as possible to the MAT, so that both tests were performed equivalently except for the HSI measurement, and (2) to develop a new method for everyday clinical use. This method should be as simple, fast, and objective as possible, so further, time-consuming follow-up measurements are not necessary. This study also intended to investigate the feasibility of perfusion analysis of the hand in combination with MAT for the first time, so that this work is only a proof of concept. Nevertheless, especially in patients with impaired perfusion, the assessment of tissue perfusion in later recordings would be of interest, since it could be analyzed whether reperfusion occurs later, but still fully, or whether tissue damage would be expected due to permanently impaired perfusion. This extension offers potential for further studies and, of course, for use in everyday clinical practice (see lines 447–462).
Comment #2:
With the HSI-parameter an additional (uncomplete) differentiation between superficial (STO2, THI) and deep perfusion (NIR) is provided and opens up the possibility to analyze perfusion problems in a more differentiated fashion. Could this be used for instance with respect to ischemic complications?
Answer #2:
Combinations of values with the possibility of drawing conclusions about tissue perfusion are as follows:
High THI and low StO2: venous congestion
Low THI and low StO2: arterial occlusion
Low THI and high StO2: following anastomosis
Low StO2 and high NIR: deep tissue perfusion is given, whereas superficial layers are undersupplied and
High StO2 and low NIR: critical situation as superficial supply can clinically hide saturation problems in deeper tissue layers.
(Already described by our group f. e. Thiem et al., 2021, Hyperspectral analysis for perioperative perfusion monitoring—a clinical feasibility study on free and pedicled flaps; Goetze et al., 2020, Identification of cutaneous perforators for microvascular surgery using hyperspectral technique - A feasibility study on the antero-lateral thigh and Grambow et al., Hyperspectral imaging for monitoring of perfusion failure upon microvascular anastomosis in the rat hind limb)
Reviewer 2 Report
This study compares HSI with Allen's test to assess collateral perfusion. HSI as a non-invasive technique shows the potential to differentiate between occlusion and perfusion, and can represent an interesting tool to verify the performance of the MAT and to confirm the diagnosis.
Comments
- I suggest publication after minor spelling corrections (graph/paragraph format, as well as text structure can be improved). The study is clinically relevant and present an innovative application of HSI in assessing perfusion. The experimental setup, analysis and results are also well presented.
- I encourage to strengthen the benefit of HSI, especially in the results discussion. I would support the discussion with a brief summary of HSI-based experiment performances compared with the state-of-the-art approach.
Author Response
This is the revision of the manuscript entitled “Is hyperspectral imaging suitable for assessing collateral circulation prior radial forearm free flap harvesting? Comparison of hyperspectral imaging and conventional Allen’s test“
Comment #1
I suggest publication after minor spelling corrections (graph/paragraph format, as well as text structure can be improved). The study is clinically relevant and present an innovative application of HSI in assessing perfusion. The experimental setup, analysis and results are also well presented.
Answer:
Spell checks were done.
Comment #2
I encourage to strengthen the benefit of HSI, especially in the results discussion. I would support the discussion with a brief summary of HSI-based experiment performances compared with the state-of-the-art approach.
Answer:
We have amended the text in accordance with the recommendation (see lines 409–439).